# Reproducibility of knee extensor and flexor contraction velocity in healthy men and women assessed using tensiomyography: A registered report

**Georg Langen** [1,2]*, **Christine Lohr** [3¤], **Olaf Ueberschär** [4,5], **Michael Behringer** [1]

**1** Department of Sports Sciences, Goethe University Frankfurt, Frankfurt am Main, Germany, **2** Department of Strength, Power and Technical Sports, Institute for Applied Training Science, Leipzig, Germany, **3** Independent Researcher, Hamburg, Germany, **4** Department of Engineering and Industrial Design, Magdeburg-Stendal University of Applied Sciences, Magdeburg, Germany, **5** Department of Biomechanics and Sport Technology, Institute for Applied Training Science, Leipzig, Germany

¤ Current address: Osteopathiepraxis, Hamburg, Germany
* langen@sport.uni-frankfurt.de

This is a Registered Report and may have an associated publication; please check the article page on the journal site for any related articles.

## Abstract

### Background

Tensiomyography measures the radial displacement of a muscle during an electrically evoked twitch contraction. Different concepts to determine the rate of displacement (Vc) from the maximum twitch exist, but information on their reproducibility is scarce. Further, different inter-stimuli intervals during progressive stimulation are used, but the effect of different intervals on Vc is unclear.

### Objectives

The first aim of this study was to investigate the within and between-day reliability of the five most frequently used Vc concepts. The second aim was to investigate the effect of different inter-stimuli intervals on Vc.

### Methods

On two consecutive days, we determined Vc of the biceps femoris long head and rectus femoris of twenty-four healthy subjects. The maximum displacement was determined twice within three minutes on day one and a third time 24 h later. Also, on day two, we applied three blocks of ten consecutive stimuli at a constant intensity of 50 mA, separated by 3 min each. Inter-stimuli intervals in randomly ordered blocks were 10 s, 20 s or 30 s, respectively.

### Results

All Vc concepts displayed good to excellent relative (ICC 0.87–0.99) and generally good absolute within- and between-day reliability for both muscles. Across Vc-concepts, absolute reliability was higher for the rectus femoris (CV% 1.3–7.95%) compared to the biceps femoris (CV% 6.06–15.30%). In both muscles, Vc was generally not affected by different inter-

**Data Availability Statement:** The preregistration including the accepted study protocol and Supporting information as well as the data and code used for analysis are openly available on the Open Science Framework platform at https://doi.org/10.17605/OSF.IO/4DU2J.

**Funding:** The author(s) received no specific funding for this work.

**Competing interests:** The authors have declared that no competing interests exist.

stimuli intervals. For most Vc concepts, repeated stimulation induced an increase regardless of the inter-stimuli interval, but this effect was mainly trivial and small at most.

## Conclusions

The reproducibility of Vc concepts was generally good but varies between different muscles. A rest interval of 10 s seems preferable to longer intervals for less time required per measurement. Following this initial study, the effect of different inter-stimuli intervals on Vc should be further investigated.

## Introduction

A muscle twitch is the contractile response to a single electrochemical signal of the nervous system. As such, a twitch provides information on muscle contractile properties and the functioning of the excitation-contraction coupling process. Tensiomyography (TMG) is a mechanomyographic method to assess skeletal muscle contractile properties [1]. TMG measures the radial displacement of a muscle belly during an electrically stimulated twitch response using a high-precision digital transducer (4 μm) positioned with a standardized pretension perpendicular to the muscle surface [2, 3]. From the radial displacement curve, spatial and temporal parameters are derived. The two most frequently reported parameters are the maximum displacement (Dm) and the contraction time (Tc) [1]. Dm provides information on muscle stiffness and atrophic [4, 5] or hypertrophic changes [6] in muscle mass and architecture. Tc refers to the time interval between 10% and 90% of Dm and is positively correlated to the proportion of slow-twitch fibres (r = 0.76 to 0.90 [7], r = 0.93 [8]) and the proportion of myosin heavy chain I (r = 0.878) [9]. Therefore, a shorter Tc is commonly associated with a higher contraction velocity [10–13]. However, it should be noted that the studies mentioned above did not report the relationship between Tc and the proportion of fast-twitch fibre types.

TMG is an involuntary method that requires no effort from the subject. Therefore, it is frequently used to assess muscular function following fatiguing exercise [14–18] or the effectiveness of different recovery strategies [19–22]. Muscular fatigue causes a slowing of muscle contraction velocity, reversing as the muscle recovers from fatigue [23–26]. Consequently, an increase in Tc of a fatigued muscle is expected which is reversed as the muscle recovers. However, previous studies found no change [16] or even a decrease of Tc [18, 27] after fatiguing exercise. Furthermore, several authors pointed out that changes in Dm inevitably lead to changes in Tc [27, 28]. Accordingly, changes in Tc should not be interpreted independently of changes in Dm.

As a result, an increasing number of studies report the rate of displacement, represented by the slope of the radial displacement curve, commonly referred to as contraction velocity (Vc). For example, in a study by Loturco et al. [29], decrements in Vc were associated with reductions in linear and change-of-direction sprint velocities after eight weeks of soccer training. Further, in female rugby players, Vc was associated with the peak power output during a 30 s Wingate test [30]. In another study, Vc of the vastus medialis muscle has been shown to decline with increasing age [31]. Regarding sex differences, Vc was lower in the lumbar erector spinae muscle [32] and the biceps femoris muscle [33] but higher in the vastus medialis muscle of women than men. Also, several previous studies have shown that Vc was decreased after fatiguing resistance exercise [14, 16, 27, 34]. In contrast, Vc was statistically not significantly changed after a simulated duathlon [35] and even significantly increased after a long-distance triathlon world championship race [36]. Thus, as the physiological determinants of Vc are not fully understood, more research is needed in this regard.

Furthermore, several concepts exist to determine Vc for different time intervals of the twitch contractions phase but there is no consensus on the most suitable approach [37]. In addition, there is a lack of studies investigating the reproducibility of these concepts, as only few studies investigated the reliability of Vc so far. Lohr et al. [38] investigated the within and between-day reliability of Vc, calculated as Vc = 0.8*Dm / Tc, in the lumbar erector spinae muscle of healthy females and males. Vc in this study showed excellent relative (intraclass correlation coefficient ICC > 0.90) and absolute (coefficient of variation, i.e., CV%, < 8%) reliability. De Paula Simola et a. [39] investigated the between-day reliability of Vc in twenty male sports students using the following concepts: $V_{10} = 0.1*Dm / Td$, where Td refers to the time interval from the electrical stimulus until 10% of Dm, and $V_{90} = 0.9*Dm / (Td + Tc)$ [39]. Absolute reliability was higher in rectus femoris (RF) and biceps femoris (CV% < 10%) compared to gastrocnemius lateralis (CV% = 12.3% and CV% = 11.3% for $V_{10}$ and $V_{90}$, respectively) [39]. Both concepts showed excellent relative reliability in all three muscles (ICC > 0.90) [39]. In addition to the concepts used in the two studies mentioned, a recent systematic scoping review of methodical approaches to determine Vc identified seven additional Vc concepts that have been used in available studies to date [37]. However, among the 62 included studies in this review, no study other than those mentioned before investigated the reproducibility of other Vc concepts. In contrast, several studies have demonstrated the adequate reproducibility of generic TMG parameters, e.g. Dm, Tc [40–46]. As such, the reproducibility of Vc-concepts calculated based on these parameters may be expected to be adequate as well. However, as Vc concepts are frequently used to assess changes in contractile properties [14, 18, 29, 34], it is essential to be able to differentiate real change from random variation. Therefore, information on the reproducibility or, more specifically, the magnitude of the typical error of commonly used Vc concept is needed. Moreover, as there is no consensus on determining Vc, a direct comparison of different Vc concepts in terms of their reproducibility within the same sample is required.

In addition to the different concepts used to determine Vc, variations within current methodical approaches exist regarding the electrical stimulation procedure to determine the radial displacement curve from which Vc is calculated. In a systematic review about the reliability of TMG, Martín-Rodríguez et al. [1] found that previous studies mainly used an inter-stimulus interval (ISI) of 10 s or 15 s to avoid fatigue or post-tetanic potentiation. However, they also stated a lack of studies investigating the optimal rest interval between consecutive stimuli [1]. To our knowledge, only two studies have investigated the effect of different ISI on tensiomyographic parameters so far. One study investigated the effect of different ISI (30 s, 10 s, 5 s) on Dm and Tc of the RF muscle during ten consecutive stimuli at a constant intensity (50 mA) [47]. The results showed that Tc was not significantly affected by different ISI, whereas Dm was significantly increased using an ISI of 10 s compared to 30 s [47]. Notably, Tc decreased significantly during ten consecutive stimuli, irrespective of the ISI, whereas Dm was unaffected [47]. However, Latella et al. [48] found no significant effect of ISI (10 s, 20 s) on Dm or Tc for the biceps brachii muscle. Consequently, they concluded that an ISI of 10 s was sufficient during five consecutive stimuli [48]. However, both studies mentioned above did not report Vc, so the effect of ISI during repeated stimulation on Vc is still unclear.

Therefore, the first aim of this study was to investigate the within and between-day reliability of the five most frequently applied concepts to calculate Vc. The second aim was to investigate the effect of different ISI during repeated stimulation on Vc, assessed by the example of biceps femoris and rectus femoris. Regarding the effect of different ISI on Vc, we hypothesized that Vc would be affected by changing ISI during ten repeated stimuli at a constant stimulation intensity.

## Materials and methods

### Study design

This study was a single group reliability study with repeated measurements within two consecutive days. The design of this study followed the recommendations for conducting and reporting diagnostic reliability studies provided in the quality appraisal of reliability studies (QAREL) checklist [49] and the guidelines for sex and gender equity in research (SAGER) [50]. Further, a completed Strengthening the reporting of observational studies in epidemiology (STROBE) checklist [51] is provided as part of the (S1 Checklist).

### Ethical approval, registration, and data availability

This study was conducted in accordance with the Declaration of Helsinki [52]. The local Ethics Committee approved this study (reference number: ER_2021.29.03_2). Participants were informed about all relevant aspects of this study and gave their written consent before their enrolment. The study protocol [53] of this registered report was preregistered after acceptance by the journal and before the recruitment and data collection begun.

### Participants and setting

We included 24 women and men in this study who were required to meet the following inclusion criteria: healthy, aged between 18–40 years, physically active for a minimum of three times per week. According to the World Health Organization, physical activity was defined as any body movement produced by skeletal muscles, including activities related to transportation, leisure time or work [54]. Exclusion criteria were the following: pregnancy, history of neuromuscular or musculoskeletal disorders, pain, or injury in the lower limbs during the last six months, previous surgical treatment to the lower limbs, practising sport on a professional level, taking prescribed medication, nontolerance or any contraindication to electrical stimulation using self-adhesive electrodes, and wearing an implanted medical device. Participants were recruited at the Institute for Applied Training Science and the Sport Sciences Department of the University of Leipzig from early Mai to the end of June 2022. During the same period, all data were collected at the Institute for Applied Trainings Science in Leipzig. In addition to anthropometric data, we recorded the subjects' physical activity level using the short form of the International Physical Activity Questionnaire (IPAC-SF) [55] presented in Table 1. During and after data collection, only the first author of this study (GL) had access to information that could be used to identify individual participants.

**Table 1. Anthropometric data and measures of physical activity level of included participants.**

| | | All (n = 24) | Women (n = 12) | Men (n = 12) |
|---|---|---|---|---|
| Age (years) | mean ± sd | 25.5 ± 3.4 | 25.9 ± 2.9 | 25.0 ± 4.0 |
| Body mass (kg) | | 71.6 ± 10.2 | 64.9 ± 8.8 | 78.3 ± 6.4 |
| Time spent on vigorous PA (min/week) | median (q1, q3) | 240.0 (131.2, 450.0) | 240.0 (112.5, 345.0) | 232.5 (168.8, 450.0) |
| Time spent on moderate PA (min/week) | | 210.0 (146.2, 360.0) | 255.0 (131.2, 390.0) | 210.0 (172.5, 360.0) |
| Time spent walking (min/week) | | 130.0 (86.2, 247.5) | 210.0 (101.2, 375.0) | 120.0 (60.0, 150.0) |
| Time spent sitting (min/week) | | 3360.0 (2415.0, 4200.0) | 3360.0 (2310.0, 3885.0) | 3255.0 (2415.0, 4200.0) |
| Energy expended on PA (MET-min/day) | | 500.4 (381.5, 861.8) | 501.0 (381.3, 861.8) | 490.7 (386.0, 794.1) |

PA: Physical activity; q1: 25th percentile; q3: 75th percentile.

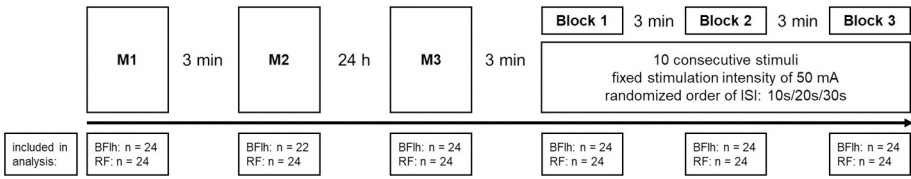

**Fig 1. Schematic illustration of the study flow.** M1, M2, M3: Measurement time points 1,2,3, respectively, mA: milliamperes, ISI: Inter-stimuli interval. *Two measurements were removed from the data for all analyses regarding M1 and M2 by listwise deletion. In both cases, Dm was unexpectedly identified as a second higher peak than the first peak of the respective displacement curve despite similar shapes of the displacement curves and first peak's amplitudes at both time points (M1 and M2). This led to outliers in the difference scores between M1–M2 of several variables.

## Experimental approach

We determined the absolute and relative within- and between-day reliability of the five most frequently used concepts to determine Vc of the biceps femoris long head (BFlh) and RF as assessed via TMG. We also determined the effect of three different rest intervals between ten consecutive stimuli at a constant intensity on these Vc concepts. Fig 1 schematically illustrates the study flow.

All tensiomyographic measurements were performed by the same researcher with more than three years of experience in performing TMG measurements. The individual maximum radial displacement curve of both the RF and BFlh was determined two times on the first day (M1, M2), separated by a three-minute pause, and a third time on the second day (M3), 24 h after the first measurement. On the second day and three minutes after the third measurement, we applied three blocks of ten consecutive stimuli each. The rest intervals between consecutive stimuli in each block were 10 s, 20 s and 30 s, respectively. The rest period between block 1, block 2 and block 3 were three minutes, respectively. We randomized the order of the three blocks with different ISI for each participant. A three-minute pause was chosen according to previous studies on the reliability of TMG parameters. The results of both studies showed that TMG parameters could be reproduced with acceptable overall reliability when a three-minute pause was given between repeated measurements. Therefore, and to foster the comparability of these previous results with ours, we applied the same pause between repeated measurements.

On the first day, we familiarized all participants with the electrical stimulation procedure by applying two stimuli with a duration of 1 ms each at 20 mA and 30 mA to the RF, followed by a three-minute rest before starting the actual measurement [47]. All measurements were taken at the same time of the day, and a constant room temperature of 21 ± 1°C. According to a recent study by Domaszewski et al. [56], orally administered caffeine can affect contractile parameters assessed by TMG. Consequently, participants were asked to refrain from caffeine intake for 2 h preceding all measurements and to avoid alcohol consumption and fatiguing exercise for 24 h before the start and during the trial to counteract possible confounding. Further, to prevent potential confounding by variations in hydration [57] and in line with a previous study [28], participants were asked to record their total dietary intake during the 24 h before the first appointment and replicate their intake during the 24 h before the second visit (S1 Appendix).

## Randomization and blinding

We used the online application research randomizer [58] to conduct a block randomization procedure [59] to determine the order of ISI during repeated stimulations for each participant. The randomisation procedure was based on four blocks corresponding to the number of

possible orders of the three different ISI. Each block contained six different numbers to account for a total of 24 subjects. To blind the rater from results of previous measurements, we chose the settings of the TMG measurement software as not to display prior measurements.

## Sample size justification

Two different calculations were made to justify the sample size for this study, which we described in detail in the study protocol of this registered report [53]. Shortly, as for our first aim to investigate the reproducibility of Vc concepts, assuming the lowest ICC reported of 0.92 [39], two measurements per subject, and an alpha error level of 0.05, 18 subjects were needed to achieve a desired precision of a confidence interval of 0.15.

As for our second aim to detect an effect of changing the interstimulus interval on Vc, our calculation was based on the raw data provided by Wilson et al. [47], see S1 Data. As we assumed three groups, ten measurements per subject, an effect size $f = 3.1$ ($Vc_{10-90\%}$) and $f = 1.6$ ($Vc_{norm}$), a nonsphericity correction coefficient of 0.7 ($Vc_{10-90\%}$) and 0.6 ($Vc_{norm}$), an alpha error level of 0.01, a power of 0.95, a number of 9 ($Vc_{10-90\%}$) subjects or 18 ($Vc_{norm}$) subjects were needed, respectively. We confirmed the estimated sample size for the ANOVA by repeating the calculations for both $Vc_{10-90\%}$ and $Vc_{norm}$ using g*power v3.1.9.2 [60]. Based on these results and accounting for potential dropouts, we recruited 24 subjects.

## Experimental set-up and procedures

To perform tensiomyographic measurements, we used a TMG-S1 electrical stimulator (TMG-BMC d.o.o., Ljubljana, Slovenia), a GD30 displacement sensor (Panoptik d.o.o., Ljubljana, Slovenia) and two squared self-adhesive electrodes (50x50 mm, Axion GmbH, Leonberg, Germany). The signal of the displacement sensor was recorded using the TMG Software v3.6 (TMG-BMC d.o.o., Ljubljana, Slovenia).

All measurements were performed on the BFlh and RF of the dominant leg. The dominant leg was defined as the leg that subjects reportedly would use to shoot a ball on a target [61]. The RF measurements were performed with subjects lying in a supine position, with their arms rested aside. The knee was supported by a triangular foam pad provided by the manufacturer at an angle of approximately 60˚ of flexion according to the neutral-zero-method [47]. The BFlh measurements were performed with subjects lying in a prone position, with their arms rested aside. The ankle was supported by a semicircular foam pad, creating a knee angle of approximately 5˚ of flexion according to the neutral-zero-method [39].

The skin in the area of measurement was cleaned with an electrode contact spray (Axion GmbH, Leonberg, Germany) and dried before the positioning of the sensor and the electrodes. The position of the sensor was determined in three steps: First, the midpoint on a line between the superior border of the patella and the anterior superior iliac spine for the RF and the BFlh, the midpoint on a line between the fibula head and ischial tuberosity was determined, according to [62]. Second, the thickest part of the muscle belly in the area of the point determined during the first step was identified by inspection and palpation during a voluntary contraction. Third, if necessary, the sensor position was adjusted at the beginning of the progressive stimulation to obtain the highest radial displacement [28, 38]. Once the position of the sensor was identified, we placed the electrodes at a distance of 7 cm from each other, measured between the facing edges of the two electrodes, and a distance of 3.5 cm between the sensor and the proximal and distal electrodes, respectively [63]. The position of the sensor and electrodes was marked using a dermatologically tested pen to ensure consistent positioning during this study.

The electrical stimulation consisted of single, monophasic, square wave stimuli with a duration of 1 ms each to elicit single isometric twitches. The stimulation started at an initial stimulation amplitude of 30 mA, and was then increased in steps of 10 mA. During M1, M2 and M3, we applied progressive electrical stimuli with rest-intervals of 15 s between consecutive stimuli to obtain the individual maximal displacement. The stimulation amplitude was increased until there was no further increase of the maximum radial displacement or until the stimulator's maximum output (110 mA) was reached. During the three blocks of repeated stimulation at 50 mA on the second day, the rest intervals between consecutive stimuli were 10 s, 20 s and 30 s, respectively.

From M1, M2 and M3, the two displacement curves with the highest first peak of the radial displacement curve of each measurement were averaged, respectively, and used for further analysis. From block 1, block 2 and block 3, we used every single displacement curve for further analysis.

## Vc concepts

Three generic tensiomyographic parameters and five different concepts to determine Vc from the displacement curve, are displayed in Fig 2.

The TMG software automatically calculated the following parameters (Fig 2A): maximum radial displacement (Dm, mm), delay time, which refers to the time interval between the stimulus and 10% of Dm (Td, ms) and contraction time, which refers to the time interval between 10% and 90% of Dm (Tc, ms). From these data, we determined Vc according to the five most frequently used concepts [37], as shown in Table 2.

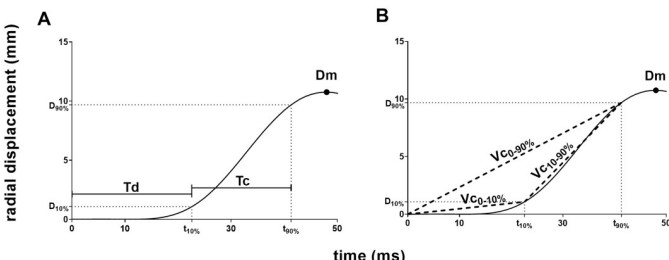

**Fig 2. Typical displacement curve during the twitch contraction phase until maximum displacement and tensiomyographic parameters.** (A) Three generic tensiomyographic parameters: maximum radial displacement (Dm), Delay time (Td), Contraction time (Tc). (B) Mean rate of displacement from electrical stimulus until 10% of Dm ($Vc_{0-10\%}$), mean rate of displacement from electrical stimulus until 90% of Dm ($Vc_{0-90\%}$) and mean rate of displacement between 10% and 90% of Dm ($Vc_{10-90\%}$).

**Table 2. Concepts applied to determine Vc with respective formulas, units, and references.**

| Vc concept | Formular | Units | Reference |
|---|---|---|---|
| mean rate of displacement from the electrical stimulus until 10% of Dm (Fig 2B) | $Vc_{0-10\%} = 0.1 \times \frac{Dm}{Td}$ | mm/s | [19] |
| mean rate of displacement from the electrical stimulus until 90% of Dm (Fig 2B) | $Vc_{0-90\%} = 0.9 \times \frac{Dm}{(Td+Tc)}$ | mm/s | [27] |
| mean rate of displacement from 10% until 90% of Dm (Fig 2B) | $Vc_{10-90\%} = 0.8 \times \frac{Dm}{Tc}$ | mm/s | [19] |
| mean rate of displacement from 10% until 90% of Dm divided by Dm | $Vc_{norm} = \frac{0.8}{Tc}$ | 1/s | [64] |
| ratio of Dm and the time interval from the stimulus until 90% of Dm (Fig 2A) | $Vc_{Dm/t90\%} = \frac{Dm}{(Td+Tc)}$ | mm/s | [29] |

## Statistical analysis

All results are reported as mean (M) and standard deviation (SD) unless differently stated. All data were checked for outliers via visual inspection of boxplots. Outliers were defined as data points outside of three times the interquartile range. The normal distribution of all variables was tested using the Shapiro-Wilk Test. To assess a systematic bias between M1-M2 and M1-M3, we performed a two-tailed paired t-test [65] or a Wilcoxon signed-rank test in case of non-normally distributed difference scores.

For the assessment of relative reliability, we calculated intra-class correlation coefficients (ICC, mean rating (k = 2), absolute agreement, two-way mixed-effects model) with 95% confidence intervals (CI) [66]. ICC scores lower than 0.5, between 0.5 and 0.75, between 0.75 and 0.9 or greater than 0.9 were interpreted as poor, moderate, good or excellent reliability [66].

Absolute reliability was quantified by calculating the standard error of measurement (SEM and SEM%), the minimal detectable change (MDC and MDC%) [67] and the CV% [65], including the respective 95% CI. The SEM represents the typical error in a measurement and was calculated as:

$$SEM = SD \times \sqrt{1 - ICC} \tag{1}$$

[65]

where SD refers to the standard deviation of the respective sample (i.e., all scores of M1 and M2 or M1 and M3, respectively). To compare the SEM of different parameters irrespectively of their units, we calculated the SEM% as:

$$SEM\% = \left(\frac{SEM}{M}\right) \times 100 \tag{2}$$

[68]

where M refers to the mean of the respective sample (i.e. all scores of M1 and M2 or M1 and M3, respectively). The MDC presents the smallest difference between two measures that can be differentiated from measurement error and hence may be considered real [67]. Following Weir et al. [67], we calculated the MDC as:

$$MDC = SEM \times 1.96 \times \sqrt{2} \tag{3}$$

[67]

The MDC% was calculated to allow a comparison between parameters independently of their units as follows:

$$MDC\% = \left(\frac{MDC}{M}\right) \times 100 \tag{4}$$

[68]

The CV% was calculated as:

$$CV\% = \frac{SD}{M} \times 100 \tag{5}$$

[65]

whereby SD refers to the standard deviation and M to the mean of individual scores in M1 and M2 or M1 and M3, respectively. A CV% > 10% was interpreted as insufficient reliability with reference to previous studies investigating the reliability of tensiomyographic parameters [38, 41, 46].

To assess the effects of different ISI on Vc during repeated stimulation, we employed a within-subject repeated-measures ANOVA. The two independent variables were ISI (10 s, 20 s, 30 s) and stimulus number (stimulus 1 to stimulus 10). The dependent variables were Dm, Td, Tc, $Vc_{0-10\%}$, $Vc_{0-90\%}$, $Vc_{10-90\%}$, $Vc_{norm}$ and $Vc_{Dm/t90\%}$ for both the BFlh and RF, respectively. Non-normally distributed data were transformed accordingly and tested again for normality. The assumption of sphericity was tested by Mauchly's test and a Greenhouse-Geisser correction applied, if necessary. Pairwise comparisons with Bonferroni correction were applied for significant main effects. The bias corrected effect size Hedge's g was calculated as:

$$Hedges's\, g_s = \frac{\overline{X_1} - \overline{X_2}}{\sqrt{\frac{(n_1-1)SD_1^2+(n_2-1)SD_2^2}{n_1+n_2-2}}} \times \left(1 - \frac{3}{4(n_1 + n_2) - 9}\right) \tag{6}$$

[69]

for pairwise comparisons between M1 and subsequent measurements during repeated stimulation. Thresholds for small, moderate, large, very large or extremely large effects will be 0.2, 0.6, 1.2, 2.0 and 4.0 [70].

Statistical significance was set at alpha ≤ 0.05. All statistical analyses were performed using R studio and respective packages.

## Results

The measurement protocol was well tolerated by the subjects and all 24 subjects fully completed the study. Descriptive statistics of all variables and mean differences between measurement time points M1 and M2 (within-day) as well as M1 and M3 (between-day) are presented in Table 3. Measures of absolute within- and between-day reliability are shown in Table 4. Fig 3 presents the ICC scores as markers of relative within- and between-day reliability. Individual and group mean data showing the effect of different ISI and repeated stimulation on all variables of the BFlh and RF are illustrated in Figs 4 and 6, respectively. Effect sizes for the differences between M1 and subsequent measurements during repeated stimulation for Dm, Td and Tc as well for Vc-concepts are presented in Figs 5 and 7, respectively.

### Within- and between-day mean differences

There were no statistically significant differences between measurements of the BFlh for any variables, except for Dm between M1 and M2 ($p = 0.047$), M1 and M3 ($p = 0.010$), respectively, and for $Vc_{10-90\%}$ between M1 and M2 ($p = 0.048$, Table 3). Regarding the RF, there were no statistically significant differences between measurements for any of the variables except for Td between M1 and M3 ($p = 0.037$, Table 3).

### Absolute reliability

Measures of absolute within and between-day reliability for both muscles are presented in Table 4. In general, the SEM% was higher between M1 and M3 compared to M1 and M2 for both muscles, indicating that absolute reliability decreased with increased time between measurements. Further, the SEM% was higher for the BFlh compared to the RF for both the

**Table 3. Tensiomyographic variables at M1, M2 and M3, as well as mean differences between measurement time points.**

| Muscle | Variable | Time points of measurement (mean ± SD), number of subjects analysed, mean difference (95% CI), p-value | | | | | | | | |
| --- | --- | --- | --- | --- | --- | --- | --- | --- | --- | --- |
| | | M1 | M2 | Δ (95% CI) | N | test statistic, p-value | M3 | Δ (95% CI) | N | test statistic, p-value |
| BFlh | Dm (mm) | 7.07 ± 2.32 | 7.34 ± 2.50 | -0.30 (-0.60–0.00) | 22 | $t(21) = -2.11$, 0.047 | 7.68 ± 2.74 | -0.60 (-1.05–-0.16) | 24 | W = 76, 0.034 |
| | Td (ms) | 30.44 ± 4.82 | 31.36 ± 5.09 | -0.56 (-1.42–0.30) | 22 | $t(21) = -1.35$, 0.192 | 30.96 ± 5.07 | -0.52 (-1.56–0.51) | 24 | $t(23) = -1.05$, 0.305 |
| | Tc (ms) | 47.41 ± 15.94 | 47.49 ± 14.15 | 1.76 (-0.87–4.39) | 22 | W = 158, 0.321 | 48.60 ± 15.69 | -1.19 (-5.19–2.81) | 24 | W = 155, 0.900 |
| | $Vc_{0-10\%}$ (mm/s) | 23.20 ± 7.49 | 23.33 ± 7.62 | -0.58 (-1.75–0.59) | 22 | $t(21) = -1.02$, 0.317 | 24.52 ± 8.29 | -1.32 (-2.79–0.16) | 24 | $t(23) = -1.85$, 0.077 |
| | $Vc_{0-90\%}$ (mm/s) | 83.44 ± 29.23 | 83.61 ± 25.79 | -4.03 (-8.17–0.10) | 22 | $t(21) = -2.03$, 0.055 | 86.54 ± 28.66 | -3.10 (-10.47–4.28) | 24 | $t(21) = -0.87$, 0.394 |
| | $Vc_{10-90\%}$ (mm/s) | 128.32 ± 52.09 | 127.20 ± 40.98 | -7.71 (-15.33–-0.08) | 22 | $t(21) = -2.10$, 0.048 | 131.74 ± 50.12 | -3.41 (-18.12–11.29) | 24 | $t(23) = -0.48$, 0.636 |
| | $Vc_{norm}$ (1/s) | 20.00 ± 10.70 | 19.25 ± 9.23 | -0.26 (-1.41–0.88) | 22 | W = 92, 0.276 | 19.19 ± 9.96 | 0.81 (-0.97–2.59) | 24 | W = 143, 0.855 |
| | $Vc_{Dm/t90\%}$ (mm/s) | 92.72 ± 32.48 | 92.90 ± 28.66 | -4.48 (-9.08–0.11) | 22 | $t(21) = -2.03$, 0.055 | 96.16 ± 31.85 | -3.44 (-11.64–4.76) | 24 | $t(23) = -0.87$, 0.394 |
| RF | Dm (mm) | 9.62 ± 2.38 | 9.79 ± 2.39 | -0.17 (-0.52–0.18) | 24 | $t(23) = -0.99$, 0.334 | 9.33 ± 2.12 | 0.29 (-0.15–0.73) | 24 | $t(23) = 1.37$, 0.184 |
| | Td (ms) | 27.56 ± 2.03 | 27.67 ± 1.93 | -0.12 (-0.48–0.24) | 24 | $t(23) = -0.69$, 0.494 | 27.93 ± 2.38 | -0.38 (-0.73–-0.02) | 24 | $t(23) = -2.21$, 0.037 |
| | Tc (ms) | 29.24 ± 3.26 | 29.36 ± 3.33 | -0.12 (-0.40–0.16) | 24 | $t(23) = -0.89$, 0.385 | 29.11 ± 3.69 | 0.12 (-0.69–0.94) | 24 | $t(23) = 0.32$, 0.755 |
| | $Vc_{0-10\%}$ (mm/s) | 34.89 ± 8.73 | 35.48 ± 9.15 | -0.58 (-2.10–0.93) | 24 | W = 134.00, 0.663 | 9.33 ± 7.21 | 1.51 (-0.29–3.32) | 24 | $t(23) = 1.73$, 0.096 |
| | $Vc_{0-90\%}$ (mm/s) | 152.91 ± 39.97 | 155.51 ± 41.96 | -2.60 (-8.40–3.20) | 24 | $t(23) = -0.93$, 0.364 | 27.93 ± 34.82 | 5.09 (-2.82–13.01) | 24 | $t(23) = 1.33$, 0.196 |
| | $Vc_{10-90\%}$ (mm/s) | 266.38 ± 74.22 | 270.80 ± 77.36 | -4.42 (-13.54–4.71) | 24 | $t(23) = -1.00$, 0.327 | 29.11 ± 68.46 | 6.30 (-8.31–20.92) | 24 | $t(23) = 0.89$, 0.382 |
| | $Vc_{norm}$ (1/s) | 27.68 ± 3.00 | 27.58 ± 3.07 | 0.10 (-0.18–0.38) | 24 | $t(23) = 0.74$, 0.465 | 33.38 ± 3.42 | -0.21 (-0.96–0.54) | 24 | $t(23) = -0.57$, 0.574 |
| | $Vc_{Dm/t90\%}$ (mm/s) | 169.90 ± 44.41 | 172.79 ± 46.62 | -2.89 (-9.34–3.56) | 24 | $t(23) = -0.93$, 0.364 | 147.81 ± 38.69 | 5.66 (-3.13–14.45) | 24 | $t(23) = 1.33$, 0.196 |

BFlh: Biceps femoris long head; RF: rectus femoris; M1, M2, M3: measurement time points 1,2,3; N: Number of subjects analysed; Δ: mean difference between respective time points; CI: confidence interval.

within- and between-day reliability (Table 4), reflecting muscle-specific variations in the typical measurement error. As for the BFlh, $Vc_{10-90\%}$ exhibited the highest and $Vc_{0-10\%}$ the lowest SEM% within Vc concepts across time points. In contrast, regarding the RF, $Vc_{0-10\%}$ displayed the highest while $Vc_{norm}$ displayed the lowest SEM% across Vc concepts between M1 and M2 and between M1 and M3, respectively.

The MDC% followed the same pattern as the SEM% with generally lower values between M1 and M2 compared to M1 and M3 for both muscles as well as lower values regarding the RF compared to the BFlh across time points (Table 4). Across Vc concepts, $Vc_{0-10\%}$ exhibited the lowest MDC% for the BFlh while $Vc_{norm}$ displayed the lowest MDC% for the RF.

Estimates of the CV% were generally lower between M1 and M2 (1.30–8.37%) as between M1 and M3 (1.92–15.30%) for both muscles, except for Tc of the RF (Table 4). Further, all variables of the BFlh displayed higher CV% estimates across time points, indicating lower absolute reliability, as compared to the RF. Between M1 and M2, CV% estimates of all variables and both muscles were below 10%, hence displaying adequate absolute reliability. However, between M1 and M3, $Vc_{0-90\%}$, $Vc_{10-90\%}$ and $Vc_{Dm/t90\%}$ of the BFlh exhibited questionable absolute reliability as the CV% estimates were 12.49%, 15.30% and 12.49%, respectively (Table 4). Across Vc concepts and time points, $Vc_{norm}$ of the BFlh and $Vc_{Dm/t90\%}$ of the RF showed the lowest CV% estimates.

**Table 4. Measures of absolute within-day and between-day reliability.**

| Muscle | Variable | M1–M2 | | | | | | M1–M3 | | | | | |
|---|---|---|---|---|---|---|---|---|---|---|---|---|---|
| | | N | SEM | SEM% | MDC | MDC% | CV% (95% CI) | N | SEM | SEM% | MDC | MDC% | CV% (95% CI) |
| BFlh | Dm (mm) | 22 | 0.34 | 4.68 | 0.93 | 12.97 | 7.54 (3.76–11.33) | 24 | 0.62 | 8.41 | 1.72 | 23.31 | 8.98 (5.71–12.25) |
| | Td (ms) | 22 | 0.98 | 3.18 | 2.73 | 8.83 | 3.03 (1.85–4.22) | 24 | 1.30 | 4.22 | 3.59 | 11.71 | 3.94 (2.49–5.39) |
| | Tc (ms) | 22 | 2.99 | 6.30 | 8.28 | 17.46 | 6.06 (3.56–8.56) | 24 | 4.95 | 10.31 | 13.72 | 28.59 | 9.30 (4.86–13.75) |
| | $Vc_{0-10\%}$ (mm/s) | 22 | 1.29 | 5.56 | 3.59 | 15.42 | 8.06 (4.32–11.80) | 24 | 1.92 | 8.05 | 5.33 | 22.32 | 9.29 (6.05–12.54) |
| | $Vc_{0-90\%}$ (mm/s) | 22 | 5.47 | 6.55 | 15.15 | 18.14 | 7.33 (4.16–10.51) | 24 | 9.07 | 10.67 | 25.14 | 29.58 | 12.49 (8.41–16.57) |
| | $Vc_{10-90\%}$ (mm/s) | 22 | 10.42 | 8.15 | 28.88 | 22.60 | 8.37 (5.22–11.52) | 24 | 18.24 | 14.03 | 50.57 | 38.89 | 15.30 (9.66–20.93) |
| | $Vc_{norm}$ (1/s) | 22 | 1.40 | 7.15 | 3.89 | 19.81 | 6.06 (3.56–8.56) | 24 | 2.05 | 10.45 | 5.67 | 28.96 | 9.30 (4.86–13.75) |
| | $Vc_{Dm/t90\%}$ (mm/s) | 22 | 6.07 | 6.55 | 16.84 | 18.14 | 7.33 (4.16–10.51) | 24 | 10.08 | 10.67 | 27.94 | 29.58 | 12.49 (8.41–16.57) |
| RF | Dm (mm) | 24 | 0.42 | 4.31 | 1.16 | 11.96 | 4.88 (2.26–7.51) | 24 | 0.54 | 5.71 | 1.50 | 15.84 | 7.24 (4.10–10.39) |
| | Td (ms) | 24 | 0.42 | 1.53 | 1.17 | 4.24 | 1.71 (1.16–2.27) | 24 | 0.45 | 1.64 | 1.26 | 4.55 | 1.92 (1.39–2.45) |
| | Tc (ms) | 24 | 0.33 | 1.12 | 0.91 | 3.10 | 1.30 (0.88–1.72) | 24 | 0.97 | 3.33 | 2.69 | 9.23 | 3.56 (2.38–4.75) |
| | $Vc_{0-10\%}$ (mm/s) | 24 | 1.80 | 5.12 | 4.99 | 14.19 | 5.59 (2.92–8.26) | 24 | 2.29 | 6.70 | 6.34 | 18.58 | 7.95 (4.91–10.98) |
| | $Vc_{0-90\%}$ (mm/s) | 24 | 6.88 | 4.46 | 19.07 | 12.37 | 5.06 (2.49–7.64) | 24 | 9.74 | 6.48 | 27.00 | 17.96 | 7.62 (4.33–10.90) |
| | $Vc_{10-90\%}$ (mm/s) | 24 | 10.81 | 4.02 | 29.96 | 11.15 | 4.84 (2.28–7.39) | 24 | 17.60 | 6.69 | 48.78 | 18.53 | 7.74 (4.12–11.35) |
| | $Vc_{norm}$ (1/s) | 24 | 0.32 | 1.17 | 0.90 | 3.25 | 1.30 (0.88–1.72) | 24 | 0.91 | 3.26 | 2.51 | 9.04 | 3.56 (2.38–4.75) |
| | $Vc_{Dm/t90\%}$ (mm/s) | 24 | 7.64 | 4.46 | 21.19 | 12.37 | 5.06 (2.49–7.64) | 24 | 10.82 | 6.48 | 30.00 | 17.96 | 7.62 (4.33–10.90) |

BFlh: Biceps femoris long head; RF: rectus femoris; N: Number of subjects analysed; SEM: Standard error of measurement; MDC: Minimal detectable change; CV: Coefficient of variation.

## Relative reliability

The relative reliability of all variables for both muscles was good to excellent between M1 and M2 and between M1 and M3 (Fig 3). As for the BFlh, lower limits of ICC score confidence intervals started at 0.87 between M1 and M2 and 0.77 between M1 and M3. All Vc concepts

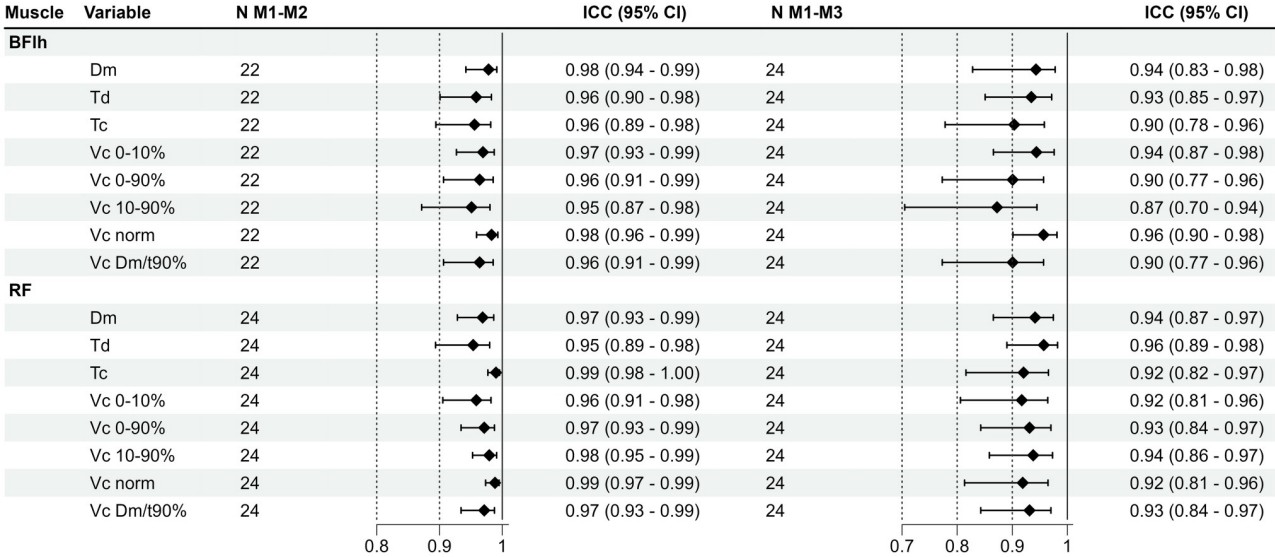

**Fig 3. Intraclass correlation coefficients (ICC) with 95% confidence intervals per muscle and measurement time point.** BFlh: Biceps femoris long head; RF: M. rectus femoris; N: Number of subjects analysed.

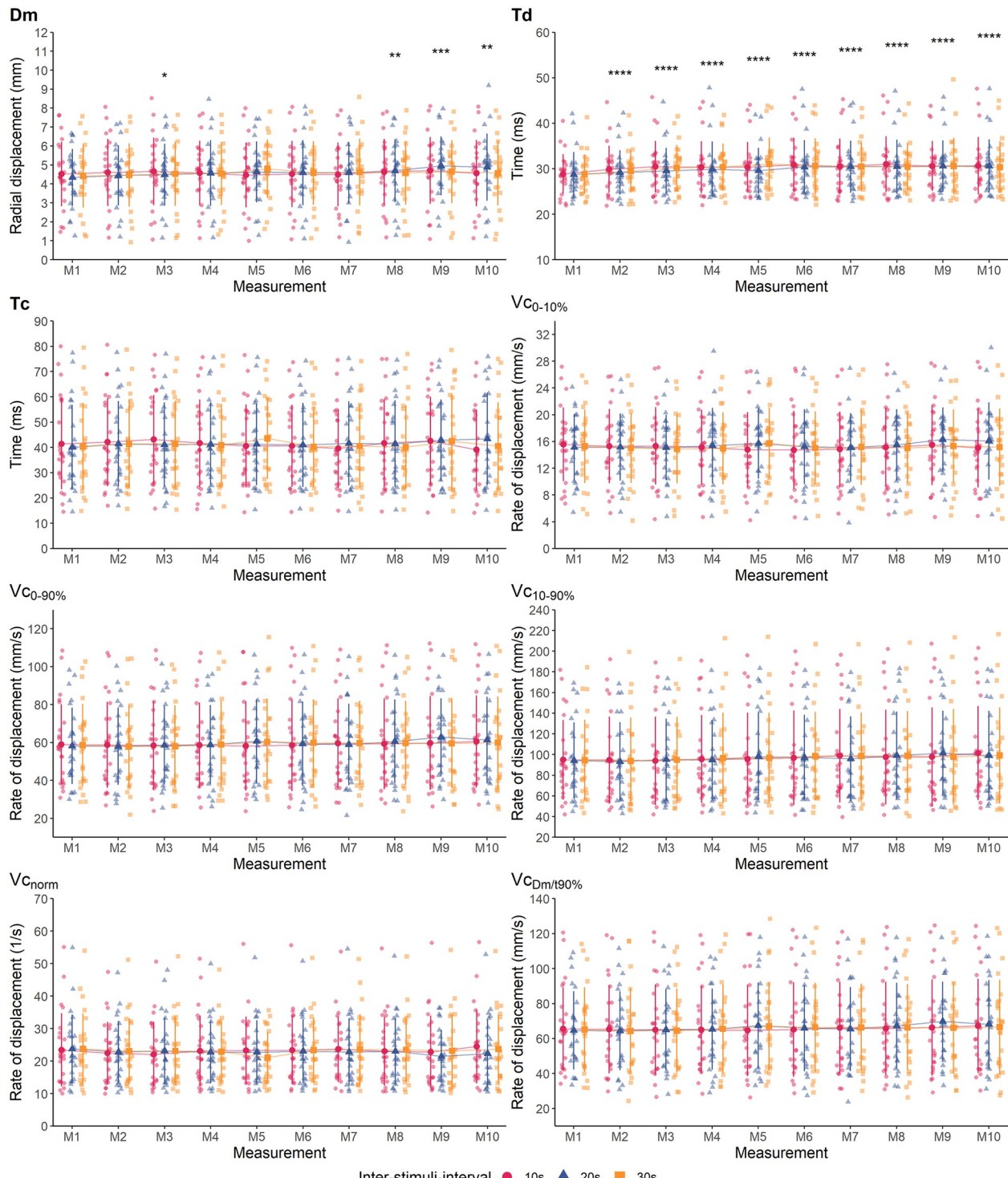

**Fig 4. Individual data points and group mean values ± standard deviation of the biceps femoris muscle per ISI and measurement time point.** Asterisks indicate significant differences compared to M1 (\*, \*\*, \*\*\*, \*\*\*\*: p < 0.05, 0.01, 0.001, 0.0001, respectively).

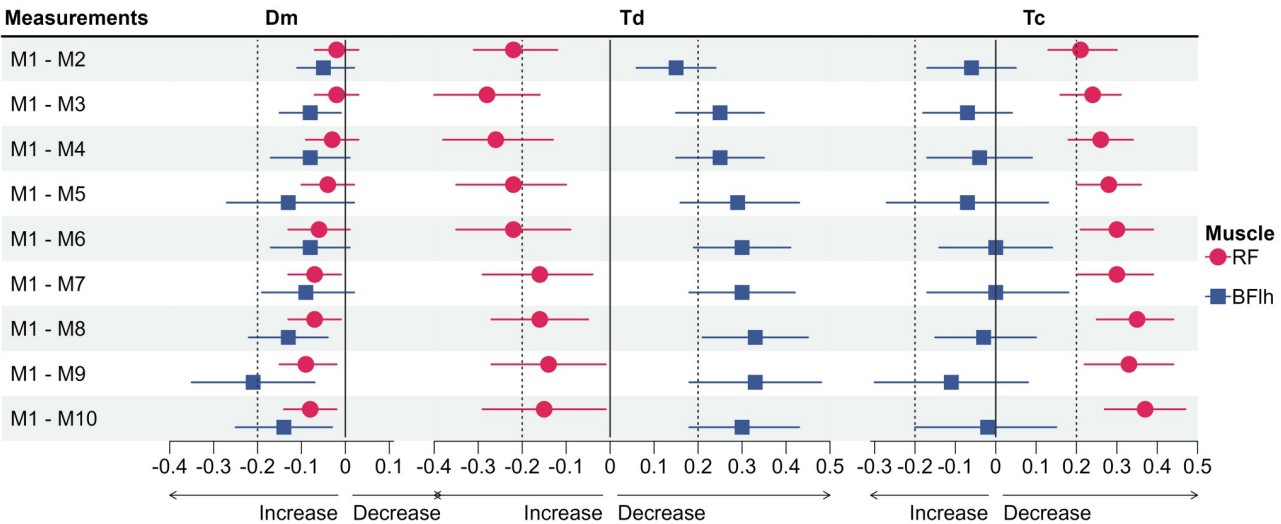

**Fig 5. Effect sizes for differences between M1 and subsequent measurements across all inter-stimuli intervals for the maximum radial displacement (Dm), delay time (Td) and contraction time (Tc).** Diamonds represent the estimates of the effect size, horizontal lines represent the confidence intervals. Horizontal lines not crossing the zero line indicate statistically significant differences. Dotted lines represent thresholds for small effects. RF: Rectus femoris; BFlh: Biceps femoris long head.

exhibited ICC estimates above 0.9, both between M1 and M2 as well as M1 and M3, except for $Vc_{10\text{-}90\%}$ at M1-M3 (0.87). Confidence intervals of ICC scores for the RF started at 0.89 and 0.81 between M1 and M2 and M1 and M3, respectively. ICC scores of all Vc concepts were above 0.9, both between M1 and M2 and M1 and M3.

### Effects of different ISI on tensiomyographic variables during repeated stimulation

There was no statistically significant interaction of ISI and stimulus number on any of the variables of the BFlh ($p > 0.05$, Table 5). Further, ISI had no statistically significant effect on any of the variables ($p > 0.05$). In contrast, except for Tc, $Vc_{0\text{-}10\%}$ and $Vc_{norm}$, repeated stimulation had a statistically significant main effect on all variables of the BFlh, which generally increased over the course of ten consecutive stimuli (Table 5). This indicates that the effect of repeated stimulation was not modulated by ISI.

For the BFlh, only Dm and Td exhibited statistically significant differences between M1 and subsequent measurements (Fig 4). As compared to M1, Dm was significantly increased in M3 ($p = 0.025$), M8 ($p = 0.001$), M9 ($p < 0.001$) and M10 ($p = 0.005$), respectively (Fig 4). However, these differences corresponded to trivial to small effects (Fig 5). Further, Td was significantly longer from M2 until M10 ($p < 0.05$) compared to M1, respectively. Yet, the corresponding effect sizes indicated only trivial to small effects of the repeated stimulation (Fig 5).

Regarding the RF, the only statistically significant interaction between ISI and Stimulus number was found for Tc ($p = 0.030$, $\eta^2 = 0.085$). Therefore, simple main effects were calculated. ISI had no significant effect on Tc at any of the then consecutive stimuli (M1 to M10 all $p > 0.05$, Table 5). In contrast, repeated stimulation had a statistically significant effect on Tc which was significantly decreased from the third stimulus with 10 s and from the second stimulus on with 20 s and 30 s of rest between consecutive stimuli ($p < 0.05$, respectively, Fig 6).

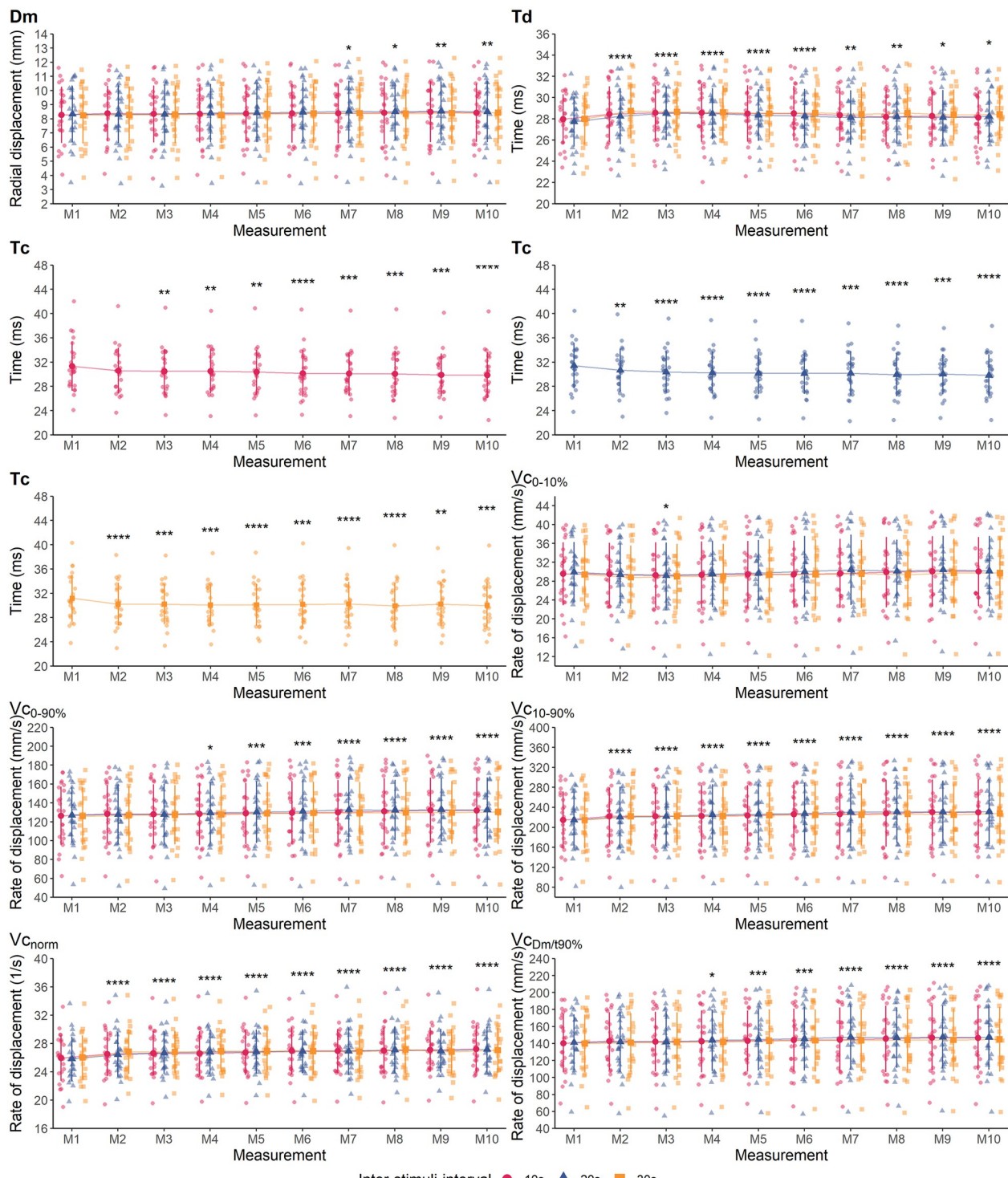

**Fig 6. Individual data points and group mean values ± standard deviation of the rectus femoris muscle per ISI and measurement.** Asterisks indicate significant differences compared to M1 (*, **, ***, ****: p < 0.05, 0.01, 0.001, 0.0001, respectively).

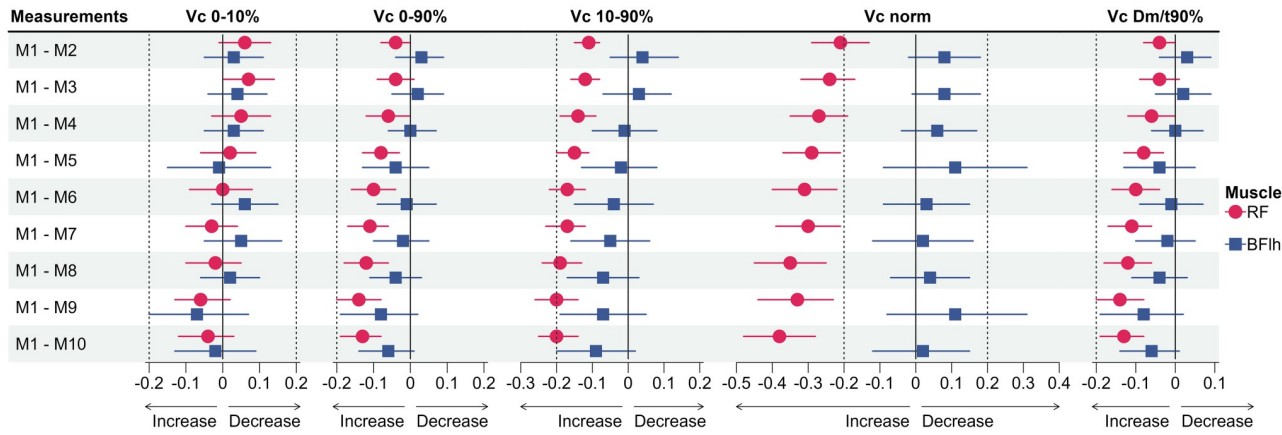

**Fig 7. Effect sizes for differences between M1 and subsequent measurements across all inter-stimuli intervals for the five different Vc concepts.**
Diamonds represent the estimates of the effect size, horizontal lines represent the confidence intervals. Horizontal lines not crossing the zero line indicate statistically significant differences. Dotted lines represent thresholds for small effects. RF: Rectus femoris; BFlh: Biceps femoris long head.

Effect sizes across ISI for the differences between M1 and subsequent measurements trended to increase towards M10 but represented still trivial to small effects (Fig 5).

Further, ISI only had a statistically significant main effect on Td ($p = 0.047$, $\eta^2 = 0.125$, Table 5). Significant differences were found during pairwise comparisons between 10 s and 30 s ($p = 0.013$, $g$ [95% CI] = -0.06 [-0.10, -0.01]) and between 20 s and 30 s ($p < 0.001$, $g = -0.10$ [-0.15, -0.06]), but not between 10 s and 20 s ($p = 0.068$, $g = 0.05$ [0.00–0.10]). However, the respective effect sizes for these comparisons represented trivial effects. Repeated stimulation also had a significant effect on Td ($p < 0.001$, $\eta^2 = 0.233$), which was significantly increased from M2 on to M10, compared to M1 ($p < 0.05$, Fig 6) but with trivial to small effect sizes (Fig 5).

Repeated stimulation had a statistically significant main effect on all remaining variables of the RF, irrespectively of the duration of ISI (Table 5). Compared to M1, Dm significantly increased from M7 on ($p < 0.05$, Fig 6), but these effects were trivial (Fig 5). $Vc_{0-10\%}$ was significantly decreased in M3 compared to M1 ($p = 0.041$), but this effect was trivial as well (Fig 7).

In contrast, $Vc_{0-90\%}$ was significantly increased from M4 on compared to M1 ($p < 0.05$). Yet, the respective effect sizes remained trivial until M10 (Fig 7). Similarly, $Vc_{10-90\%}$ increased significantly from M2 on ($p < 0.001$) but these effects were small to trivial (Fig 7). Following the same pattern, $Vc_{norm}$ significantly increased from M2 on compared to M1 ($p < 0.001$). The respective effect sizes continuously increased, reaching small effects from M5 on but remained small until M10 (Fig 7). Lastly, $Vc_{Dm/t90\%}$ was significantly increased from M4 on compared to M1 ($p < 0.05$), but with trivial effect sizes throughout (Fig 7).

## Discussion

This study compared the five most frequently used concepts to determine Vc in terms of within and between-day reliability, evaluated in the biceps femoris muscle and rectus femoris muscle. The relative within- and between-day reliability of all Vc concepts was good to excellent for both the BFlh and RF. The absolute reliability was adequate for the RF within and between days across all Vc concepts, as assessed by the CV%. For the BFlh, Vc concepts

**Table 5. Results of the within-subject repeated-measures ANOVA.**

| Muscle | Variable | N | Interaction F(df) | p | η² | ISI F(df) | p | η² | Stimulus number F(df) | p | η² |
|---|---|---|---|---|---|---|---|---|---|---|---|
| BFlh | Dm | 24 | 2.00(3.7, 84.8) | 0.107 | 0.080 | 0.11(2, 46) | 0.893 | 0.005 | 4.51(4.2, 97.3) | 0.002 | 0.164 |
| | Td | 24 | 1.14(5.7, 131.1) | 0.346 | 0.047 | 0.92(2, 46) | 0.408 | 0.038 | 11.21(3.9, 90.4) | <0.001 | 0.328 |
| | Tc | 24 | 1.06(4.5, 102.9) | 0.385 | 0.044 | 0.06(1.2, 28.2) | 0.852 | 0.003 | 0.97(4.1, 93.4) | 0.429 | 0.040 |
| | $Vc_{0-10\%}$ | 24 | 1.53(5.3, 122.8) | 0.183 | 0.062 | 0.55(2, 46) | 0.580 | 0.023 | 2.15(5.0, 113.8) | 0.065 | 0.086 |
| | $Vc_{0-90\%}$ | 24 | 1.65(5.4, 123.9) | 0.148 | 0.067 | 0.29(2, 46) | 0.752 | 0.012 | 3.94(4.3, 98.9) | 0.004 | 0.146 |
| | $Vc_{10-90\%}$ | 24 | 1.18(7.6, 175.7) | 0.317 | 0.049 | 0.06(1.5, 34.0) | 0.891 | 0.003 | 3.43(3.5, 79.5) | 0.016 | 0.130 |
| | $Vc_{norm}$ | 24 | 1.06(3.0, 69.4) | 0.371 | 0.044 | 0.16(1.3, 29.0) | 0.755 | 0.007 | 1.36(3.6, 83.1) | 0.257 | 0.056 |
| | $Vc_{Dm/t90\%}$ | 24 | 1.65(5.4, 123.9) | 0.148 | 0.067 | 0.29(2, 46) | 0.752 | 0.012 | 3.94(4.3, 98.9) | 0.004 | 0.146 |
| RF | Dm | 24 | 0.77(8.0,183.9) | 0.631 | 0.032 | 0.65(2, 46) | 0.529 | 0.027 | 4.03(2.7, 62.3) | 0.013 | 0.149 |
| | Td | 24 | 1.01(8.2, 188.2) | 0.431 | 0.042 | 3.28(2, 46) | 0.047 | 0.125 | 6.97(3.3, 75.2) | <0.001 | 0.233 |
| | Tc | 24 | 2.14(8.6, 197.8) | 0.030 | 0.085 | M1 0.60(2, 46) | 1.000 | 0.025 | 10s 14.35(3.7, 86.1) | <0.001 | 0.384 |
| | | 24 | | | | M2 2.91(2, 46) | 0.700 | 0.112 | | | |
| | | 24 | | | | M3 1.08(2, 46) | 1.000 | 0.045 | | | |
| | | 24 | | | | M4 3.11(2, 46) | 0.500 | 0.119 | | | |
| | | 24 | | | | M5 0.86(2, 46) | 1.000 | 0.036 | 20s 14.2(4.4, 101.2) | <0.001 | 0.382 |
| | | 24 | | | | M6 0.02(2, 46) | 1.000 | 0.001 | | | |
| | | 24 | | | | M7 0.22(2, 46) | 1.000 | 0.009 | | | |
| | | 24 | | | | M8 0.30(2, 46) | 1.000 | 0.013 | | | |
| | | 24 | | | | M9 1.13(2, 46) | 1.000 | 0.047 | 30s 9.96(5.2, 118.6) | <0.001 | 0.302 |
| | | 24 | | | | M10 0.42(1.6, 37.1) | 1.000 | 0.018 | | | |
| | $Vc_{0-10\%}$ | 24 | 1.00(8.2, 189.5) | 0.440 | 0.042 | 1.52(2, 46) | 0.229 | 0.062 | 5.34(3.3, 75.9) | 0.002 | 0.188 |
| | $Vc_{0-90\%}$ | 24 | 1.25(7.3, 168.0) | 0.278 | 0.051 | 1.29(2, 46) | 0.286 | 0.053 | 12.36(2.6, 58.7) | <0.001 | 0.350 |
| | $Vc_{10-90\%}$ | 24 | 1.68(7.3, 167.0) | 0.114 | 0.068 | 0.80(2, 46) | 0.455 | 0.034 | 22.76(2.3, 53.7) | <0.001 | 0.497 |
| | $Vc_{norm}$ | 24 | 2.00(8.7, 200.0) | 0.057 | 0.076 | 0.14(2, 46) | 0.869 | 0.006 | 24.90(3.4, 79.2) | <0.001 | 0.520 |
| | $Vc_{Dm/t90\%}$ | 24 | 1.25(7.3, 168.0) | 0.278 | 0.051 | 1.29(2, 46) | 0.286 | 0.053 | 12.36(2.6, 58.7) | <0.001 | 0.350 |

BFlh: Biceps femoris long head; RF: rectus femoris; N: Number of subjects analysed; F(df): F-value(degrees of freedom); p: p-value; η²: partial eta squared; ISI: Inter-stimulus-interval; M1-M10: Measurement 1 to 10, respectively.

exhibited sufficient within-day reliability but insufficient between-day reliability (i.e., CV% above 10%) in three of five concepts.

Further, this study investigated whether Vc would be affected by rest intervals of 10 s, 20 s or 30 s during ten consecutive stimuli at a constant intensity of 50 mA. Contrary to our initial hypothesis, ISI did not affect Vc of the BFlh or RF during repeated stimulation. However, repeated stimulation significantly affected the majority of Vc concepts regardless of ISI in both muscles, but these effects were trivial to small. This is the first study to investigate the effect of different ISI during repeated stimulation on the most frequently used Vc concepts.

## Reproducibility of Vc concepts and generic TMG parameters

The reproducibility of generic TMG parameters has been previously reported in a number of studies [40–45]. In addition, a recent systematic review with meta-analysis reported a high to excellent relative reliability of Td, Tc and Dm in several muscles with ICC scores of 0.91, 0.95 and 0.98, respectively, but conflicting to moderate positive results in terms of absolute reliability for these parameters [2]. Concerning the muscles investigated in this study, previous studies reported a good to excellent relative reliability (ICC scores from 0.82 to 0.99 for Td, Tc, Dm)

[39, 40, 44, 45] for the BFlh but only partially sufficient absolute reliability (CV% scores from 2.4% to 19.8% for Td, Tc, Dm) [39, 40, 45]. To the RF, ICC scores from 0.87 to 0.92 and CV% scores from 3.8% to 9.3% have been reported [39], thus representing good to excellent relative and sufficient absolute reliability of Td, Tc and Dm. Accordingly, our results (ICC scores from 0.90 to 0.99 and CV% scores from 0.42% to 9.30%) are consistent with previous findings, demonstrating excellent relative and acceptable absolute within- and between-day reliability for Td, Tc and Dm of the BFlh and RF, respectively.

However, only a few previous studies so far have investigated the reproducibility of Vc [38, 39, 71]. Lohr et al. [38] investigated the within and between-day reliability of $Vc_{10-90\%}$ in the lumbar erector spinae muscle of healthy females and males. $Vc_{10-90\%}$ in this study showed excellent relative (ICC > 0.90) and absolute (CV% < 8%) reliability. De Paula Simola et al. [39] investigated the between-day reliability of $Vc_{10\%}$ and $Vc_{90\%}$ in the RF, BFlh, and gastrocnemius lateralis of male sports students. In their study, $Vc_{10\%}$ and $Vc_{90\%}$ both showed excellent relative reliability in all three muscles (ICC > 0.90) [39]. Absolute reliability was higher in RF and BFlh (CV% < 10%) compared to gastrocnemius lateralis (CV% = 12.3% and CV% = 11.3% for $V_{10}$ and $V_{90}$, respectively) [39]. In a recent study, Mesquita et al. [71] compared six Vc concepts in the biceps brachii muscle regarding their between-day reliability at three different joint angles. Their results showed that relative and absolute reliability indices varied considerably across Vc concepts and joint angles, as ICC estimates ranged from 0.32 to 0.78 and CV% estimates ranged from 2.8% to 12.3% [71]. Our study adds to these results by informing on the within and between-day reliability of the five most frequently used Vc concepts in direct comparison. In line with the results by Lohr et al. [38] and de Paula Simola et al. [39], we found good to excellent relative within- and between-day reliability of all Vc concepts for both the BFlh and RF. The CV%, as an indicator of absolute reliability, was lower for within- than for between-day reliability for the BFlh and RF, which is consistent with the previous findings by Lohr et al. [38]. Further, the CV% was generally lower in the RF than in the BFlh, indicating, in accordance with previous studies [38, 39], that the reproducibility of Vc appears to be muscle-specific.

When comparing Vc concepts, $Vc_{norm}$ displayed the best overall reproducibility with respect to the CV% and ICC. In addition to the CV%, we reported both the absolute and relative SEM and MDC, respectively, for the Vc concepts and three generic TMG parameters (i.e., Td, Tc and Dm) of the BFlh and RF. These indices provide information on the typical error of a measure and the minimal change in the measure that may be considered real [67]. Hence, our results may serve as a reference to interpret changes in these parameters within the BFlh and RF of healthy males and females.

Further, we suggest that different Vc concepts should not be used interchangeably for two reasons: First, Vc may be influenced by different physiological factors when calculated for different time intervals of the twitch contraction phase, e.g. 0%-10% or 10%-90% of Dm, as recently pointed out [37]. Consequently, depending on the part of the contraction phase considered for the calculation, Vc would convey different information about the contractile properties of the respective muscle belly. Second, in line with the results obtained by Mesquita et al. [71], our results show that the reproducibility varies across Vc-concepts, especially in terms of the absolute reliability, as shown in Table 4.

## Effect of ISI on Vc concepts during repeated stimulation

A rest interval of 10 s to 30 s between consecutive stimuli during a TMG measurement is commonly applied [2, 37] to avoid the effect of fatigue or potentiation on the muscle twitch response [43, 46]. However, the optimal rest interval between consecutive stimuli is still

unclear [1]. Therefore, we investigated if different ISI would affect Vc during repeated stimulation. Previously, only two studies have investigated the effect of different ISI during repeated stimulation on generic TMG parameters [47, 48]. Latella et al. [48] found no significant effect of ISI (10 s, 20 s) within five consecutive stimuli on Dm or Tc of the biceps brachii muscle. Consequently, they concluded that an ISI of 10 s is sufficient to avoid the effects of fatigue or potentiation. In contrast, Wilson et al. [47] reported that Dm of the RF significantly increased when using an ISI of 10 s compared to 30 s during ten consecutive stimuli. They also reported that Tc significantly decreased within ten consecutive stimuli irrespectively of ISI (30 s, 10 s or 5 s) [47].

Following up on these two previous studies, we investigated the effect of three different ISI (10 s, 20 s, 30 s) during ten consecutive stimuli on Vc and generic TMG parameters of the BFlh and RF. According to Wilson et al. [47], a potentiation effect may explain an increase in Dm respectively a decrease in Tc during repeated stimulation. Indeed, repeated low-frequency electrical stimulation is known to induce post-tetanic potentiation [72], leading to increased peak force, decreased time to peak force and increased rate of force development during an electrically evoked twitch [73]. Thus, it is still conceivable that a potentiation effect could also lead to an increase in Vc if Dm increases even while Tc remains unchanged or vice versa, as reported by Wilson et al. [47].

However, there is a lack of research directly comparing the force or torque produced to muscle displacement during the same twitch under different physiological conditions, e.g., potentiation. Only one recent study [74] compared the torque and displacement of the same twitch after five repeated short maximal voluntary contractions intended to induce potentiation. In their study, Abazovic et al. [74] found an increased amplitude of both the twitch torque and displacement of the vastus medialis and lateralis. Further, after the potentiation intervention, the displacement-derived Tc was shortened in both muscles. However, the torque-derived Tc was longer in the VM and remained unchanged in the VL. On the one hand, these findings show that the twitch displacement may be affected by a potentiation effect. On the other hand, this effect may not necessarily match the potentiation effect on the twitch torque response.

Considering these previous results, we hypothesized that different ISI would affect Vc during repeated stimulation. However, our results showed that ISI had no significant effect on any of the parameters investigated, except for Td of the RF. Instead, repeated stimulation had a statistically significant effect on all parameters except for Tc, $Vc_{0-10\%}$ and $Vc_{norm}$ of the BFlh. In general, repeated stimulation induced an increase in the majority of Vc concepts of both muscles, indicating a potentiation effect, as suggested by Wilson et al. [47]. However, as the statistical significance does not inform on the magnitude of the effect, we also reported effect sizes for pairwise comparisons between M1 and subsequent measurements. These effect sizes indicated merely trivial to small effects of repeated stimulation on all variables and both muscles investigated. Thus, although repeated stimulation with an ISI of 10 s, 20 s or 30 s may induce a potentiation effect on Vc and generic TMG parameters, the magnitude of this effect appears to be small at most. Consequently, in line with Latella et al. [48], our results suggest that applying a longer ISI of 30 s or 20 s instead of 10 s may provide no benefit in terms of methodical validity. In contrast, an ISI of 10 s may be preferable to a longer ISI in favour of a reduced time per measurement.

## Limitations

A limitation of our study is that we only assessed two muscles of the lower extremities. Nevertheless, as our results show, the absolute reliability of Vc appears to be muscle specific.

Therefore, we suggest that future studies should investigate the reproducibility of Vc in a broader range of muscles, including muscles of the upper extremities. Another limitation of our study is that we included only healthy and physically active women and men between 18 and 40 years of age. The effect of different ISI on Vc might differ in older subjects or athletes due to age- and training-related shifts in the muscle fibre spectrum [13, 31, 75].

## Conclusions and practical applications

According to our results, the five most frequently used Vc-concepts displayed adequate overall reproducibility for the BFlh and RF. Of the Vc concepts investigated, $Vc_{norm}$ showed the highest overall reproducibility across measurement time points and muscles. However, the absolute reliability of Vc concepts was generally lower for the BFlh compared to the RF. This difference presumably results from anatomical differences (width of muscle bellies, distance to neighbouring muscle bellies) between the two muscles, which may render the measurement of the BFlh more susceptible to confounding effects from co-contracting muscle bellies. As such, our results suggest that the reproducibility of Vc concepts is muscle-specific and therefore needs to be further investigated in different muscles. Further, we reported indices of absolute reliability (SEM, MDC), which may be used as reference values to assess changes in contractile properties in the BFlh and RF and within the population from which our sample was obtained.

Our results also show that Vc and generic TMG parameters were generally not affected by different ISI of 10 s, 20 s and 30 s during repeated submaximal stimulation. Regardless of ISI, repeated stimulation resulted in an increased Vc for most Vc concepts in terms of a potentiation effect. However, this effect's magnitude was mainly trivial and small at most. Consequently, an ISI of 10 s may be prefered to a longer ISI in order to spend less time per measurement. However, our study is the first to investigate the effect of different ISI on Vc during repeated stimulation. Moreover, our results are limited to the BFlh and the RF of healthy, physically active women and men between 20 and 36 years. Therefore, we suggest that the effect of different ISI during repeated stimulation on Vc should be investigated in future studies and different populations.

## Supporting information

**S1 Checklist. STROBE statement regarding the manuscript titled "Reproducibility of knee extensor and flexor contraction velocity in healthy men and women assessed using tensiomyography: A registered report".**
(PDF)

**S1 Appendix. 24 h dietary intake protocol template.**
(PDF)

**S1 Table. Calculated rate of displacement data used for sample size estimation.**
(XLSX)

## Author Contributions

**Conceptualization:** Georg Langen.

**Data curation:** Georg Langen.

**Formal analysis:** Georg Langen, Christine Lohr.

**Investigation:** Georg Langen.

**Methodology:** Georg Langen, Christine Lohr, Olaf Ueberschär, Michael Behringer.

**Project administration:** Georg Langen.

**Software:** Georg Langen.

**Supervision:** Michael Behringer.

**Visualization:** Georg Langen.

**Writing – original draft:** Georg Langen.

**Writing – review & editing:** Georg Langen, Christine Lohr, Olaf Ueberschär, Michael Behringer.

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
