## [Decision Letter · Decision Letter 0]

19 Apr 2023

PONE-D-23-06908Reproducibility of knee extensor and flexor contraction velocity in healthy men and women assessed using tensiomyography: A registered reportPLOS ONE

Dear Dr. Langen,

Thank you for submitting your manuscript to PLOS ONE. After careful consideration, we feel that it has merit but does not fully meet PLOS ONE’s publication criteria as it currently stands. Therefore, we invite you to submit a revised version of the manuscript that addresses the points raised during the review process.

ACADEMIC EDITOR:

Dear Authors, two experts in the filed revised your manuscript reporting some issues you should consider during the revision process.

We look forward to receiving your revised manuscript.

Kind regards,

Emiliano Cè

Academic Editor

PLOS ONE

Journal Requirements:

Reviewers' comments:

Reviewer's Responses to Questions

**Comments to the Author**

1. Does the manuscript adhere to the experimental procedures and analyses described in the Registered Report Protocol?

If the manuscript reports any deviations from the planned experimental procedures and analyses, those must be reasonable and adequately justified.

Reviewer #1: Yes

Reviewer #2: Yes

2. If the manuscript reports exploratory analyses or experimental procedures not outlined in the original Registered Report Protocol, are these reasonable, justified and methodologically sound?

A Registered Report may include valid exploratory analyses not previously outlined in the Registered Report Protocol, as long as they are described as such.

Reviewer #1: Yes

Reviewer #2: Yes

3. Are the conclusions supported by the data and do they address the research question presented in the Registered Report Protocol?

The manuscript must describe a technically sound piece of scientific research with data that supports the conclusions. The conclusions must be drawn appropriately based on the research question(s) outlined in the Registered Report Protocol and on the data presented.

Reviewer #1: Yes

Reviewer #2: Yes

4. Have the authors made all data underlying the findings in their manuscript fully available?

Reviewer #1: Yes

Reviewer #2: Yes

5. Is the manuscript presented in an intelligible fashion and written in standard English?

Reviewer #1: Yes

Reviewer #2: Yes

6. Review Comments to the Author

Please use the space provided to explain your answers to the questions above. (Please upload your review as an attachment if it exceeds 20,000 characters)

Reviewer #1: The present article aims to test the reproducibility of the rate of displacement (Vc) and to test different inter-stimuli intervals to ascertain the effects on Vc. This parameter has some controversy because there are several formulas to calculate its value since its conception in the late 1990s. Until now, there is no article that has verified which formula is better to analyze Vc and why. In fact, the authors of this article recently published (2022) an interesting methodological review in this regard.

This type of methodological study is necessary to clarify the usefulness and reproducibility of these types of markers that are used above all in the clinical field and in the field of sports performance. Overall, the study is methodologically well carried out and a lot of attention has been paid to write everything in a precise and concrete way, which is to be appreciated. Therefore, I want to highlight only a few minor changes:

- Introduction: (p. 3, L-61) add information about what the TMG is, since the authors started talking about the TMG and it is taken for granted that the readers know that it is.

- Introduction: (p.3, L67) add the Pearson value corresponding to the positive correlation mentioned.

- Introduction: (p.5, L117-118) there are 2 objectives and only one hypothesis. The hypothesis is not related to objective one or two, so it seems to respond to a third objective. Better clarify this. An objective, a hypothesis.

- Methods: please, when referring to BF in the whole text, specify always that it is the biceps femoris long head (BFlh), which is the portion of the BF that I think it was measured by reading the methods.

- Methods: (p. 7, L178) why 3 minutes and no more or less than that; I want to know if it is a random number or if there is any article that had studied the influence of rest periods.

- Methods: (p. 8, L193) no reason to capitalize "research randomizer", change it.

Reviewer #2: There are still many issues that need to be addressed regarding TMG, hence the relevance of the aim of study. Yet, the justification of the study does not seem to be strong enough. It has been verified that when the evaluator is truly an expert, the intra-day and inter-day reliability of Dm and Tc measurements is adequate. If Vc relates both variables, it is not a surprise that it has also a good reliability, as it has been verified in some other studies. So, what does this work really contribute differently? I think this question needs to be answered in a much deeper way. The absence of many papers should not be the main reason.

Line 67. As far as I know, Tc has only been associated with a greater or lesser number of slow fibers, never with IIa or IIx fibers. Please be precise, it cannot be established that the contraction speed is high with this exclusive relationship regarding slow-twitch fibers. Some of the articles used to justify that statement do not seem to point it out such thing. In addition, linking the transverse radial displacement time of the MTU with the contraction speed does not seem adequate, or at least it is far from having been demonstrated.

Line 72-73. This statement is speculative, since Vc does not really measure muscle contraction velocity, it measures the transverse radial muscle belly displacement at a certain milliampere, that is, of type I fibers. In fact, it has been pointed out that Vc would not be a suitable concept and has been proposed in numerous works Vrd (radial displacement velocity). In this sense, how could it be explained that Vrd increases after performing a long duration iroman? This is the case of the recent work by Cuba-Dorado et al. (2022) (neuromuscular changes after a long distance triathlon word championship). Briefly, it is simply a ratio of the transverse radial displacement of the UMT.

Line 78-79. In the recent work of Mesquita et al. (2023) (Contraction velocity of the elbow flexors assessed by TMG) compares the results obtained by different formulas and their reliability, indicating that they are not interchangeable formulas. What do you think about this?

Line 246. This incremental protocol is very common in TMG, but there are some issues to clarify. As far as I know, the fact that at a certain amplitude of the stimulus (mA) there is no increase in Dm, does not mean that it does not occur with a greater amplitude. This represents a certain bias, that is, the plateau does not remain constant at all subsequent amplitudes. To make sure you get the maximum radial displacement it is necessary to use a protocol until maximum output. What do you think about this aspect?

Line 274. The statistical treatment is well resolved.

Line 481. To the best of my knowledge, in the recent work by Cuba-Dorado et al. (2022) also reported the reproducibility of Vrd or Vc (an inappropriate term as I have previously pointed out), obtaining good indicators with elite and well-trained triathletes.

Line 495. What do you think this difference between the two muscles is due? It would be appropriate to explain this statement, since it seems that MDC is very different depending on the muscle tested.

Please provide a practical applications section for all those who use TMG to measure the contractile properties of muscles.

Please limit your conclusions to the sample obtained. Can this be extrapolated to well-trained athletes aged 18-30?

7. PLOS authors have the option to publish the peer review history of their article (what does this mean?). If published, this will include your full peer review and any attached files.

Reviewer #1: **Yes: **Saul Martin Rodriguez

Reviewer #2: No

---

## [Author Response · Author response to Decision Letter 0]

13 May 2023

PONE-D-23-06908

Reproducibility of knee extensor and flexor contraction velocity in healthy men and women assessed using tensiomyography: A registered report

We thank both reviewers for providing valuable comments and suggestions to help us improve the manuscript. We have responded to all comments below and revised the manuscript accordingly. We appreciate your further review of the manuscript. We also hope that you will find that our responses are adequate and that the revised manuscript can be accepted for publication.

Responses to Reviewer #1: (Changes to the manuscript have been highlighted in yellow)

The present article aims to test the reproducibility of the rate of displacement (Vc) and to test different inter-stimuli intervals to ascertain the effects on Vc. This parameter has some controversy because there are several formulas to calculate its value since its conception in the late 1990s. Until now, there is no article that has verified which formula is better to analyse Vc and why. In fact, the authors of this article recently published (2022) an interesting methodological review in this regard.

This type of methodological study is necessary to clarify the usefulness and reproducibility of these types of markers that are used above all in the clinical field and in the field of sports performance. Overall, the study is methodologically well carried out and a lot of attention has been paid to write everything in a precise and concrete way, which is to be appreciated. Therefore, I want to highlight only a few minor changes:

Thank you for your positive feedback, which we highly appreciate.

- Introduction: (p. 3, L-61) add information about what the TMG is, since the authors started talking about the TMG and it is taken for granted that the readers know that it is.

Thank you for this remark. We have added the following information:

"Tensiomyography (TMG) is a mechanomyographic method to assess skeletal muscle contractile properties [1]. TMG measures the radial displacement of a muscle belly during an electrically stimulated twitch response using a high-precision digital transducer (4 µm) positioned with a standardised pretension perpendicular to the muscle surface [2,3]." (please see page 3, lines 62 to 66)

- Introduction: (p.3, L67) add the Pearson value corresponding to the positive correlation mentioned.

Thank you for your comment. We have added the Pearson values. The sentence now reads:

"Tc refers to the time interval between 10 % and 90 % of Dm and is positively correlated to the proportion of slow-twitch fibres (r = 0.76 to 0.90 [7], r = 0.93 [8]) respectively the proportion of myosin heavy chain I (r = 0.878) [9]." (please see page 3, lines 71 to 72)

- Introduction: (p.5, L117-118) there are 2 objectives and only one hypothesis. The hypothesis is not related to objective one or two, so it seems to respond to a third objective. Better clarify this. An objective, a hypothesis.

Thank you for this remark. We changed this part as follows:

"Regarding the effect of different ISI on Vc, we hypothesised that Vc would be affected by changing ISI during ten repeated stimuli at a constant stimulation intensity." (page 6, lines 142 to 144).

- Methods: please, when referring to BF in the whole text, specify always that it is the biceps femoris long head (BFlh), which is the portion of the BF that I think it was measured by reading the methods.

Thank you for pointing this out. You are right and we changed this throughout the whole text accordingly, including all tables and figures (for example, please see page 2, line 42, page 8, line 186 or page 28, line 622).

- Methods: (p. 7, L178) why 3 minutes and no more or less than that; I want to know if it is a random number or if there is any article that had studied the influence of rest periods.

Thank you for this question. To the best of our knowledge, no study has investigated the influence of rest periods on the within-day reliability of TMG parameters. 

However, there is a previous study by Lohr et al. on the reproducibility of Dm, Tc and Vc of the lumbar erector spinae muscle (doi: 10.1007/s00421-018-3867-2), in which a 3-minute rest between repeated measurements was performed. We decided to apply the same rest interval to enhance the comparability of their results and ours. Further, in a study by Tous-Fajardo et al. 2010 (doi: 10.1016/j.jelekin.2010.02.008) on the inter-rater reliability of TMG measurement and the effect of inter-electrode distance on TMG parameters, a 3-minute rest period between measurements was applied as well. As both studies showed that TMG parameters could be reproduced with overall acceptable reliability, we felt 3 minutes would also be a reasonable rest period for our study.

We have also added this information to the manuscript. (please see page 8 lines 206 to 211)

- Methods: (p. 8, L193) no reason to capitalise "research randomiser", change it.

Thank you for your note. We have changed this line accordingly (page 9, line 225).

Responses to Reviewer #2: (Changes have been highlighted in blue)

There are still many issues that need to be addressed regarding TMG, hence the relevance of the aim of study. Yet, the justification of the study does not seem to be strong enough. It has been verified that when the evaluator is truly an expert, the intra-day and inter-day reliability of Dm and Tc measurements is adequate. If Vc relates both variables, it is not a surprise that it has also a good reliability, as it has been verified in some other studies. So, what does this work really contribute differently? I think this question needs to be answered in a much deeper way. The absence of many papers should not be the main reason.

We thank you for your constructive feedback on our manuscript. According to your comment, we have changed the end of the second paragraph of the introduction as follows:

"In addition to the concepts used in the two studies mentioned, a recent systematic scoping review of methodical approaches to determine Vc identified seven additional Vc concepts that have been used in available studies to date [37]. However, among the 62 included studies in this review, no study other than those mentioned before investigated the reproducibility of other Vc concepts. In contrast, several previous studies have demonstrated the adequate reproducibility of generic TMG parameters, e.g. Dm, Tc [40–46]. As such, the reproducibility of Vc-concepts calculated based on these parameters may be expected to be adequate as well. However, as Vc concepts are frequently used to assess changes in contractile properties [14,18,29,34], it is essential to be able to differentiate real change from random variation. Therefore, information on the reproducibility or, more specifically, the magnitude of the typical error of commonly used Vc concept is needed. Moreover, as there is no consensus on determining Vc, a direct comparison of different Vc concepts in terms of their reproducibility within the same sample is required." (please see pages 4 to 5, lines 110 to 122)

In this regard, our results provide information on the typical error of the five most commonly used Vc concepts for two of the most commonly studied muscles in a sample of healthy women and men, as mentioned on page 24 in lines 534 to 539 and on page 27 in lines 612 to 615.

Further, our study is the first to report the effect of different inter-stimuli intervals on Vc. As we have mentioned on page 5 in lines 127 to 128 and 137 to 138, information on the optimal rest interval between repeated stimuli during TMG measurements is lacking. To the best of our knowledge, only two studies have investigated the effect of different inter-stimuli intervals on Dm and Tc but not on Vc. Hence, our results provide novel, practically relevant evidence regarding the effect of different rest intervals between consecutive stimulation on generic TMG parameters and commonly applied Vc-concepts.

Line 67. As far as I know, Tc has only been associated with a greater or lesser number of slow fibers, never with IIa or IIx fibers. Please be precise, it cannot be established that the contraction speed is high with this exclusive relationship regarding slow-twitch fibers. Some of the articles used to justify that statement do not seem to point it out such thing. In addition, linking the transverse radial displacement time of the MTU with the contraction speed does not seem adequate, or at least it is far from having been demonstrated.

Thank you for your remark on this. We have changed this sentence as follows:

"Tc refers to the time interval between 10 % and 90 % of Dm and is positively correlated to the proportion of slow-twitch fibres (r = 0.76 to 0.90 [7], r = 0.93 [8]) and the proportion of myosin heavy chain I (r = 0.878) [9]. Therefore, a shorter Tc is commonly associated with a higher contraction velocity [10–13]. However, it should be noted that the studies mentioned above did not report the relationship between Tc and the proportion of fast-twitch fibre types. (please see page 3, lines 70 to 75)

Line 72-73. This statement is speculative, since Vc does not really measure muscle contraction velocity, it measures the transverse radial muscle belly displacement at a certain milliampere, that is, of type I fibers. In fact, it has been pointed out that Vc would not be a suitable concept and has been proposed in numerous works Vrd (radial displacement velocity). In this sense, how could it be explained that Vrd increases after performing a long duration iroman? This is the case of the recent work by Cuba-Dorado et al. (2022) (neuromuscular changes after a long distance triathlon word championship). Briefly, it is simply a ratio of the transverse radial displacement of the UMT.

Thank you for your remark. We would like to respectfully point out that, on page 4, in lines 84 to 85, we do not state that Vc would measure the muscle contraction velocity. However, to make this differentiation more explicit to the reader, we have rewritten the respective paragraph, which now reads as follows:

"As a result, an increasing number of studies report the rate of displacement, represented by the slope of the radial displacement curve, commonly referred to as contraction velocity (Vc). For example, in a study by Loturco et al., decrements in Vc were associated with reductions in linear and change-of-direction sprint velocities after eight weeks of soccer training [29]. Further, in female rugby players, Vc was associated with the peak power output during a 30 s Wingate test [30]. In another study, Vc of the vastus medialis muscle has been shown to decline with increasing age [31]. Regarding sex differences, Vc was lower in the lumbar erector spinae muscle [32] and the biceps femoris muscle [33] but higher in the vastus medialis muscle of women than men. Also, several previous studies have shown that Vc was decreased after fatiguing resistance exercise [14,16,27,34]. In contrast, Vc was statistically not significantly changed after a simulated duathlon [35] and even significantly increased after a long-distance triathlon world championship race [36]. Thus, as the physiological determinants of Vc are not fully understood, more research is needed in this regard.

Furthermore, several concepts exist to determine Vc for different time intervals of the twitch contractions phase but there is no consensus on the most suitable approach [37]. […]" (please see page 4 lines 84 to 98).

Line 78-79. In the recent work of Mesquita et al. (2023) (Contraction velocity of the elbow flexors assessed by TMG) compares the results obtained by different formulas and their reliability, indicating that they are not interchangeable formulas. What do you think about this?

Thank you for this comment. We completely agree with the statement you mentioned. To make this more clear to the reader, we have added the following sections to the Discussion section:

"In a recent study, Mesquita et al. [66] compared six Vc concepts in the biceps brachii muscle regarding their between-day reliability at three different joint angles. Their results showed that relative and absolute reliability indices varied considerably across Vc concepts and joint angles, as ICC estimates ranged from 0.32 to 0.78 and CV% estimates ranged from 2.8 % to 12.3 % [66]." (please see page 24, lines 520 to 524)

"Further, we suggest that different Vc concepts should not be used interchangeably for two reasons: First, Vc may be influenced by different physiological factors when calculated for different time intervals of the twitch contraction phase, e.g. 0 %-10 % or 10 %-90 % of Dm, as recently pointed out [29]. Consequently, depending on the part of the contraction phase considered for the calculation, Vc would convey different information about the contractile properties of the respective muscle belly. Second, in line with the results obtained by Mesquita et al. [66], our results show that the reproducibility varies across Vc-concepts, especially in terms of the absolute reliability, as shown in Table 4." (please see page 25, lines 540 to 547)

Line 246. This incremental protocol is very common in TMG, but there are some issues to clarify. As far as I know, the fact that at a certain amplitude of the stimulus (mA) there is no increase in Dm, does not mean that it does not occur with a greater amplitude. This represents a certain bias, that is, the plateau does not remain constant at all subsequent amplitudes. To make sure you get the maximum radial displacement it is necessary to use a protocol until maximum output. What do you think about this aspect?

Thank you for your comment and your question. It certainly is possible that Dm could continue to increase if the stimulation was further increased after a plateau in Dm had been reached. Consequently, this fact should be recognised as a limitation of an incremental stimulation protocol. However, strictly speaking, even if the maximum stimulation intensity of the TMG stimulator was consistently applied, this potential bias still could not be ruled out as a plateau in Dm may be reached at an intensity above the maximum output. Accordingly, the TMG stimulator's maximum power output represents a limitation, which has been recognised in several previous studies (Garcia-Garcia et al. 2019: https://doi.org/10.2147/OAJSM.S161485 ; Latella et al. 2019: https://doi.org/10.1016/j.jelekin.2019.02.002 ; Langen et al. 2022: https://doi.org/10.1016/j.jelekin.2022.102702).

However, we designed our measurement protocol based on the most frequently reported practices in studies that have investigated Vc (Langen et al. 2022: https://doi.org/10.1016/j.jelekin.2022.102702).) so that our results can be applied to commonly used practices. Therefore, we decided to define the endpoint of the stimulation procedure as described on page 11, lines 277 to 279. 

To our knowledge, there is no study comparing different stimulation procedures in terms of their validity and reproducibility, which would certainly be a valuable contribution to developing a standardised TMG measurement procedure.

Line 274. The statistical treatment is well resolved.

Thank you for your positive feedback, which is well appreciated.

Line 481. To the best of my knowledge, in the recent work by Cuba-Dorado et al. (2022) also reported the reproducibility of Vrd or Vc (an inappropriate term as I have previously pointed out), obtaining good indicators with elite and well-trained triathletes.

Thank you for the comment and the reference to the very interesting article. This article represents a valuable contribution to understanding the effect of muscular fatigue after prolonged submaximal loading under actual race conditions on tensiomyographically assessed contractile properties of elite athletes. However, in contrast to the three studies we have cited on page 24 in lines 512 to 513, it appears that the reproducibility of Vc or Vrd was not the primary focus of the study you mentioned, as it reports the pooled reliability indices of two muscles in the method sections.

Line 495. What do you think this difference between the two muscles is due? It would be appropriate to explain this statement, since it seems that MDC is very different depending on the muscle tested.

Please provide a practical applications section for all those who use TMG to measure the contractile properties of muscles.

Thank you for this comment. To increase the practical applicability of our results, we have changed the section header and have rewritten the whole section as follows:

"Conclusions and practical applications

" According to our results, the five most frequently used Vc-concepts displayed an adequate overall reproducibility for the BFlh and RF. Of the Vc concepts investigated, Vcnorm showed the highest overall reproducibility across measurement time points and muscles. However, the absolute reliability of Vc concepts was generally lower for the BFlh compared to the RF. This difference presumably results from anatomical differences (width of muscle bellies, distance to neighbouring muscle bellies) between the two muscles, which may render the measurement of the BFlh more susceptible to confounding effects from co-contracting muscle bellies. As such, our results suggest that the reproducibility of Vc concepts is muscle-specific and therefore needs to be further investigated in different muscles. Further, we reported indices of absolute reliability (SEM, MDC), which may be used as reference values to assess changes in contractile properties in the BFlh and RF and within the population from which our sample was obtained. 

Our results also show that Vc and generic TMG parameters were generally not affected by different ISI of 10 s, 20 s and 30 s during repeated submaximal stimulation. Regardless of ISI, repeated stimulation resulted in an increased Vc for most Vc concepts in terms of a potentiation effect. However, this effect's magnitude was mainly trivial and small at most. Consequently, an ISI of 10 s may be prefered to a longer ISI in order to spend less time per measurement. However, our study is the first to investigate the effect of different ISI on Vc during repeated stimulation. Moreover, our results are limited to healthy, physically active women and men between 20 and 36 years. Therefore, we suggest that the effect of different ISI during repeated stimulation on Vc should be investigated in future studies and different populations." (please see page 26, lines 603 to 625)

Please limit your conclusions to the sample obtained. Can this be extrapolated to well-trained athletes aged 18-30?

Thank you for pointing this out. According to your comment, we have limited our conclusions to our sample and the muscle investigated. The respective section now reads as follows:

"Further, we reported indices of absolute reliability (SEM, MDC), which may be used as reference values to assess changes in contractile properties in the BFlh and RF and within the population from which our sample was obtained." (please see page 27, lines 612 to 615).

"Moreover, our results are limited to the BFlh and the RF of healthy, physically active women and men between 20 and 36 years. Therefore, we suggest that the effect of different ISI during repeated stimulation on Vc should be investigated in future studies and different populations." (please see page 28, lines 622 to 625)

---

## [Decision Letter · Decision Letter 1]

4 Jul 2023

Reproducibility of knee extensor and flexor contraction velocity in healthy men and women assessed using tensiomyography: A registered report

PONE-D-23-06908R1

Dear Dr. Langen,

We’re pleased to inform you that your manuscript has been judged scientifically suitable for publication and will be formally accepted for publication once it meets all outstanding technical requirements.

Kind regards,

Emiliano Cè

Academic Editor

PLOS ONE

Additional Editor Comments (optional):

Reviewers' comments:

Reviewer's Responses to Questions

**Comments to the Author**

1. Does the manuscript adhere to the experimental procedures and analyses described in the Registered Report Protocol?

If the manuscript reports any deviations from the planned experimental procedures and analyses, those must be reasonable and adequately justified.

Reviewer #1: Yes

2. If the manuscript reports exploratory analyses or experimental procedures not outlined in the original Registered Report Protocol, are these reasonable, justified and methodologically sound?

A Registered Report may include valid exploratory analyses not previously outlined in the Registered Report Protocol, as long as they are described as such.

Reviewer #1: Yes

3. Are the conclusions supported by the data and do they address the research question presented in the Registered Report Protocol?

The manuscript must describe a technically sound piece of scientific research with data that supports the conclusions. The conclusions must be drawn appropriately based on the research question(s) outlined in the Registered Report Protocol and on the data presented.

Reviewer #1: Yes

4. Have the authors made all data underlying the findings in their manuscript fully available?

Reviewer #1: Yes

5. Is the manuscript presented in an intelligible fashion and written in standard English?

Reviewer #1: Yes

6. Review Comments to the Author

Please use the space provided to explain your answers to the questions above. (Please upload your review as an attachment if it exceeds 20,000 characters)

Reviewer #1: The authors have provided a satisfactory response to my inquiries, addressing my requests adequately. I have no further additions to make at this time.

7. PLOS authors have the option to publish the peer review history of their article (what does this mean?). If published, this will include your full peer review and any attached files.

Reviewer #1: **Yes: **Saúl Martín Rodríguez

---

## [Editor Report · Acceptance letter]

21 Jul 2023

PONE-D-23-06908R1 

Reproducibility of knee extensor and flexor contraction velocity in healthy men and women assessed using tensiomyography: A registered report 

Dear Dr. Langen:

I'm pleased to inform you that your manuscript has been deemed suitable for publication in PLOS ONE. Congratulations! Your manuscript is now with our production department. 

Kind regards, 

on behalf of

Prof. Emiliano Cè 

Academic Editor

PLOS ONE